# The Origin, Function, Distribution, Quantification, and Research Advances of Extracellular DNA

**DOI:** 10.3390/ijms232213690

**Published:** 2022-11-08

**Authors:** Kaixin Yang, Lishuang Wang, Xinghong Cao, Zhaorui Gu, Guowei Zhao, Mengqu Ran, Yunjun Yan, Jinyong Yan, Li Xu, Chunhui Gao, Min Yang

**Affiliations:** 1Key Laboratory of Molecular Biophysics of the Ministry of Education, College of Life Science and Technology, Huazhong University of Science and Technology, Wuhan 430074, China; 2State Key Laboratory of Agricultural Microbiology, College of Resources and Environment, Huazhong Agricultural University, Wuhan 430070, China

**Keywords:** extracellular DNA, origin, functions, distribution, organisms, extraction, quantification

## Abstract

In nature, DNA is ubiquitous, existing not only inside but also outside of the cells of organisms. Intracellular DNA (iDNA) plays an essential role in different stages of biological growth, and it is defined as the carrier of genetic information. In addition, extracellular DNA (eDNA) is not enclosed in living cells, accounting for a large proportion of total DNA in the environment. Both the lysis-dependent and lysis-independent pathways are involved in eDNA release, and the released DNA has diverse environmental functions. This review provides an insight into the origin as well as the multiple ecological functions of eDNA. Furthermore, the main research advancements of eDNA in the various ecological environments and the various model microorganisms are summarized. Furthermore, the major methods for eDNA extraction and quantification are evaluated.

## 1. Introduction

A general consensus has risen that the total DNA in a natural environment is comprised of intracellular DNA (iDNA) and extracellular DNA (eDNA). DNA has been thought to exist only inside the cell for long periods of time. However, an increasing amount of evidence has shown the presence of DNA in the extracellular space, and these eDNA outside the cell may account for a large proportion of the total DNA [1]. Compared with iDNA that is located in living cells, eDNA is outside the cell, and it is widely distributed in soil, sediments, feces and aquatic ecosystems and other ecological niches, and it has also been discovered in tissue cultures and in the blood of the human body and some other animals [2,3,4,5]. Moreover, the ratio of eDNA:iDNA could provide valuable information about the microbial activity in different environments [6,7], although the activity of different microorganisms is not completely correlated with the ratio [8]. These results indicate that eDNA has the potential to be a proxy for specific microbial activities [9].

Initially, eDNA was thought to be derived from lysed cells. Nowadays, it has been well documented that both the lysis-dependent and lysis-independent pathways were involved in eDNA release in prokaryotic and eukaryotic cells [10]. There are two general forms of eDNA, which are single-stranded DNA (ssDNA) and double-stranded DNA (dsDNA), and the latter accounts for a significant portion of the total eDNA in the environment due to the instability of ssDNA [11]. The increasing amount of evidence has shown that eDNA plays a significant role in multiple biological processes such as biofilm formation and maturation, horizontal gene transfer (HGT), and multiple element cycling including for carbon, nitrogen, and phosphate [12,13,14,15]. However, the biological roles of eDNA are different across different species, and they even vary with the different life stages in the same organism. For example, in *Neisseria meningitidis*, eDNA has different roles in early and late biofilm formation [16].

During the past decades, biodiversity is disappearing at an alarming rate, and the large-scale assessment and monitoring of biodiversity has become a priority. Therefore, eDNA-based biomonitoring has emerged as an effective tool for detecting biodiversity changes at a large scale, and it offers us a better understanding of the microbial community composition as well as the community function [9].

So far, most of the available reviews on eDNA focus on one specific habitat and one specific organism. Considering this, the aim of this review is to discuss the role of eDNA in the various ecological environments and different species. In addition, we attempt to explore the origin and the function of eDNA. Given that only a small part of viable microorganisms can be cultured through traditional culturing methods [17], studying eDNA could provide essential information of the genetic and functional diversity of those uncultured microorganisms and the historical information about past climatic conditions [18]. In addition, this review presents a summary of eDNA extraction and quantification methods.

## 2. The Origin of eDNA

The mechanism of eDNA release varies in different species. Generally, two different pathways are involved in the origin of eDNA; one is the lysis-dependent pathway, and the other one is lysis-independent pathway (Figure 1).

### 2.1. Lysis-Dependent Pathway of eDNA Release

The lysis-dependent pathway tends to be accompanied with cell lysis, and usually is induced by some lethal factors (Figure 1) such as bacterial endolysin [19,20], prophage [21], virulence factors [19,22] or antibiotics [23], thus, leading to the release of cell contents including DNA. For example, prophage endolysin encoded within the R- and F-pyocin gene cluster can stimulate eDNA release through explosive cell lysis in *Pseudomonas aeruginosa* (*P. aeruginosa*) [20]. In addition, pyocyanin promotes eDNA release through H_2_O_2_-mediated cell lysis in *P. aeruginosa* [24]. The λSo prophage induction that is triggered by iron is able to accelerate the eDNA release from the lysed cells [25]. Virulence factors, including hemo-lysins, leukotoxins lipases and autolysin, and autolysin are the core of quorum sensing which controls cell lysis [26,27]. Cell lysis could be promoted by a high concentration of different monovalent cations [28]. Additionally, the mechanical disruption and pathogenic bacteria invasion of plants can cause the enzymatic degradation of the plant structure, thus resulting in eDNA release from plants [15].

### 2.2. Lysis-Independent Pathway of eDNA Release

eDNA can also be actively released into the natural environments. It can be detected and isolated from a number of bacteria during their growth in vitro [29,30,31,32,33,34]. In addition, eDNA can be secreted via membrane vesicles (MVs), eosinophils, and mast cells [35,36]. The eDNA derived from MVs plays a critical role in biofilm formation and maturation as a component of the biofilm matrices in *Streptococcus mutans* [35]. Neutrophil extracellular traps (NETs), which are highly ordered biofilm components consisting of protein and DNA and the sticky matrix around the cell, have been reported to be the main source of eDNA in biofilm structures [37]. eDNA can be released from different kinds of living cells in response to a pathogen infection. For instance, neutrophils secrete eDNA and other components to form a NET in the immune system of the human body to escape from the pathogen invasion [2,38,39,40]. Most root tips of plants can release eDNA, which exerts the same function as eDNA in human NETs [36,41].

## 3. Ecological Functions of eDNA

### 3.1. eDNA Serves as an Essential Structural Component of Biofilm Matrix

Biofilm is a microbial aggregate that is formed by microorganisms in order to distinguish themselves from the external environment and maintain their own stability (Figure 2). It includes cells and extracellular polymeric substances (EPS) [42]. At first, EPS was thought to be mainly composed of polysaccharides. With further research, EPS has been reported that contained an abundant amount of protein and eDNA. eDNA is the important component of EPS, and it participates in regulating the formation of biofilms and maintaining the dimensional stability of biofilms [26]. After their adherence of bacteria onto the surface of the soil particle, the bacterium begins to proliferate and secrete EPS. The eDNA interacts with other components in the EPS to form a firm structure, thus, maintaining the spatial structure and stability of the biofilm [43]. One previous study has shown that after adding DNaseI to the biofilm, the EPS components would be digested, and similar results have also been observed after adding proteinase [44]. This phenomenon indicates that both eDNA and protein are integral to the structure of the biofilm and eDNA interacts with proteins in the biofilm. In addition, eDNA can also interact with extracellular polysaccharides [45]. The polysaccharide PsI and eDNA can form a PsI-eDNA fiber network, which acts as the skeleton of the biofilm and provides support for the bacteria [46].

### 3.2. eDNA Acts as Genetic Information Carrier in HGT

HGT is essential in bacterial evolution (Figure 2) [47,48]. Mobile genetic elements such as plasmids, transposons, integrons, genomic islands could act as the carriers of HGT [49,50,51,52]. The size of the eDNA fragment is an important factor in HGT, mainly because only small eDNA fragments can transport into cell membrane [53]. The eDNA contained in bacterial biofilms is also one of mobile genetic elements and it provides rich resources for naturally occurring genetic transformation and phage-induced gene transfer [54]. The efficiency of biofilm-mediated gene transfer is improved. It has been reported that bacterial viability is closely related to eDNA release and subsequent biofilm formation. Moreover, it has been reported that inactivated bacteria addition can promote HGT in a way that is relative to the DNA addition [55].

### 3.3. eDNA Serves as a Main Organic Compound in Nutrient Cycle

Phosphorus, as a key element, participates in critical biogeochemical reactions in the life process, and it has a great impact on cell growth [56]. Previous studies have envisioned that eDNA in marine sediments may serve as an important P resource for cell growth and turnover [57]. In order to explore the potential applications of eDNA, researchers have made an assessment of the daily P requirement and P supply, and the results have shown that eDNA might supply 41% of the bacterial requirement for P in marine ecosystem, which offers new insight into the role of eDNA in the P cycle [58]. eDNA can serve not only as the source of phosphorus, but also as that of carbon and nitrogen for different species in various environments (Figure 2). For example, the genus *Shewanella* can use eDNA as the sole source of P, carbon, and energy to maintain normal growth [59]. It has been reported that *Escherichia coli* and *Haemophilus influenzae* could take advantage of their natural ability to take in eDNA as food for competitive survival in the case of nutritional deficiency [60,61].

## 4. Distribution of eDNA in Different Environments

eDNA is widely distributed in the soil, sediments, feces, and other environments (Figure 3).

### 4.1. eDNA in Soil

eDNA widely exists in soil, and it accounts for about 40% of the total DNA pool. Its content of eDNA varies significantly in different soil regions, types, and layers, and most of the eDNA is concentrated in the upper layers of the soil [15]. The range of eDNA content in soil is about 0.03–200 μg/g [53]. At the increasing depth of the soil layer, the eDNA content tends to decrease, and its reduction is much more obvious than that of organic carbon [1]. In addition, it has been found that the higher the fertilizer supply is, then the lower the eDNA concentration is [62]. It is estimated that more than 70% of the DNA molecules present in soil are derived from fungi [63].

The eDNA in soil can be divided into two fractions with one that is tightly bound to the soil particles and the other that is weakly bound [64] so that the eDNA can be protected against degradation by nucleases in the soil, thus, stably persisting in the soil for a long time [53]. Therefore, eDNA can serve as a historical microbial gene reservoir, reflecting the diachronic biodiversity of the investigated environments. The long-term stable existence of eDNA in soil can lead to the accumulation of exogenous genes such as antibiotic resistance genes (ARGs) which may be passed between cells, meanwhile soil microorganisms acquire ARGs through homologous recombination, thus, potentially generating new resistance strains, especially resistant pathogenic bacteria. These strains have strong resistance and are not easily subjected to environmental stresses, which pose great threats to the environments [65]. In addition, eDNA occupies a large proportion (about 10%) of the soil P pool, which enables the microbes to take up DNA as a nutrition source of carbon, nitrogen, phosphorus, and nucleic acid precursors [66]. Surprisingly, adding eDNA to a culture medium has been reported to promote the growth of the lateral roots and root hairs in *Arabidopsis* [67]. However, it has been found that conspecific eDNA treatment can inhibit the growth of plants and the soil microbe, whereas heterologous eDNA treatment cannot, indicating that this inhibitory effect might be due to the maintenance of the microbial diversity [68]. However, more research is needed to further explore the underlying mechanism. In addition, plants can release NETs in which eDNA is able to enhance the resistance of root tip to soil pathogens, but the resistance of these pathogens would be lost with a DNaseI treatment [69].

### 4.2. eDNA in Sediments

eDNA can be found in various types of sediments, such as marine sediments, river sediments and freshwater sediments. Among them, the marine sediments are the largest sediment eDNA reservoir. Increasing evidence suggests that more than 90% of DNA in sediments is extracellular, and it has been estimated that the content of eDNA ranges from 0.30 to 0.45 Gt in deep-sea sediments [70]. The content of eDNA varies in the different sediments. A study of Haihe River sediments has demonstrated that the concentration of eDNA (96.8 ± 19.8 μg/g) is much higher that of iDNA (76.7 ± 13.0 μg/g) [71]. In ferruginous sediments from Lake Towuti of Indonesia, the eDNA concentration is at around 0.5–0.6 μg/g in the surface layer of the sediments, and the concentration of eDNA has the same trends in the sediments and soil, namely, it decreases with the increasing depth [72]. Apart from the eDNA concentration, the eDNA fragment size also decreases as the depth increases [5].

The most of the eDNA in the sediments is bound or adsorbed to particles to fight against nuclease degradation [73]. Additionally, the eDNA in sediments possesses multiple roles. For example, the eDNA in the marine sediments can not only reflect the biogenesis processes on different time scales, but also provide information on biodiversity and genetic diversity of different ecosystems in both ancient and modern times [74]. In addition, eDNA could be used to obtain past climate and environmental information through a taxa analysis, and it promises to be a very powerful tool for predicting future environmental changes and the function of the ecosystem [75].

### 4.3. eDNA in Feces

eDNA can be extracted from cattle feces, and its content is about 202.4 μg/g from fresh excrements [76]. Additionally, the content of eDNA in swine mature ranges from 9.6 to 9.7 μg/g dw (dry weight) [77]. There findings indicate that eDNA is an essential ARGs reservoir. eDNA receives increasing attention since the ARG pool in mature animals poses a threat to the ecological environments and human health. Livestock and poultry manure is generally considered to be a natural host for ARGs, and the environment exposure of manure from farms may drive the circulation of ARGs [78]. In humans, gut microbiome dysregulation may lead to the occurrence of a series of diseases, and thus, extracellular viral-like particles (eVLPs) and virus DNA from human feces could be extracted for metagenome and morphology analyses [79,80].

### 4.4. eDNA in Other Ecological Environments

eDNA exists in various liquid environments such as seawater, freshwater, river water, and lake water. The eDNA concentration in aquatic environments can reach up to 88 μg/L [81]. The concentration of eDNA is always associated with the trophic status and the season with its concentrations ranging from 2.5 to 46 μg/L in mesotrophic water, while it ranged from 11.5 to 72 μg/L in eutrophic water [82].

## 5. Research Advances of eDNA in Model Organisms

Although it is believed that eDNA in the natural environments comes from a variety of living organisms, the related research on eDNA function is still limited to typical model organisms. Therefore, we introduce the related progress on this by taking *Bacillus*, *Pseudomonas*, and *Escherichia* as examples.

### 5.1. eDNA in Bacillus

The active release of eDNA from living and intact cells in *Bacillus subtilis* can be observed in liquid cultures [83]. The mechanism of active eDNA release is positively correlated with an evolutionary advantageous behavior since the evolutionary function of eDNA is always related to HGT [84]. Moreover, the unpurified extracellular plasmid DNA can facilitate the gene delivery in HGT [85]. It is well known that the ability of strains to take up of exogenous DNA is always linked to the growth state and the growth environments [86]. It has been displayed that as an effective antimicrobial lipopeptide, surfactin can promote the HGT in *Bacillus subtilis* by enhancing the eDNA release.

eDNA plays an important role in the structural formation of biofilms in *Bacillus*, and it can act as an adhesin in the biofilm formation of *Bacillus cereus* [87]. The latest research has shown that there is an interaction between eDNA with exopolysaccharide in modulating 3D biofilm architecture in the early stages of *Bacillus subtilis* biofilm development [88]. In addition, the DNA–amyloid complexes are essential for the adherence and aggregation of a *Bacillus licheniformis* biofilm, allowing this strain to be adapted to different growth conditions [89]. eDNA can promote the adsorption of *Sulfobacillus thermosulfidooxidans* on chalcopyrite surface, thereby increasing the copper extraction [90].

### 5.2. eDNA in Pseudomonas

The genus *Pseudomonas* is widespread in various environments, such as soil, water, plants, and clinical samples, and the most common strains mainly include *P. aeruginosa*, *Pseudomonas fluorescens*, *Pseudomonas putida*, and *Pseudomonas syringae* [91]. There is increasing interest in the opportunistic pathogenic bacterium *P. aeruginosa* due to its ability to form biofilms, and the formation of these biofilms leads to chronic infections owing to high levels of drug resistance [92,93]. A microscopic observation has demonstrated that eDNA is mainly located in the stalks of mushroom-shaped multicellular structures [94]. eDNA can interact with other components of EPS, and it plays multiple roles in *Pseudomonas* biofilm formation. For example, the exopolysaccharide Psl/Pel–eDNA complex can form the biofilm skeleton to structurally support in *P. aeruginosa* [46,95]. eDNA is found to be required for the early stage of the biofilm formation of *P. aeruginosa* after DNaseI treatments [96]. Additionally, the acidification of biofilm by eDNA and the chelation of eDNA and cations, respectively, induce aminoglycoside resistance and antibiotic resistance in their resistance to antibiotic stress in *P. aeruginosa* [97,98]. eDNA is a proinflammatory factor of the biofilm in activating neutrophil [99]. It is exciting that eDNA has great potential in the bioremediation of heavy metal-contaminated soil due to its capability of chelating cations, and eDNA can facilitate chromium adsorption and complexation in *Pseudomonas putida* [100]. Furthermore, the content of eDNA increases obviously under Cu^2+^ stress in unsaturated *Pseudomonas putida* CZ1 biofilms, and eDNA mainly binds copper ions at the first coordination site Cu-O or Cu-N [101].

Cell lysis and cell death can promote eDNA release, and the R- and F-pyocin gene cluster can stimulate Lys-mediated explosive cell lysis in *P. aeruginosa* [20]. In addition, three holins including AlpB, CidA, and Hol participate in Lys-mediated eDNA release [19].

In *P. aeruginosa*, eDNA release may be activated by *Pseudomonas* quinolone signal (PQS), pyocyanin, and lambda prophage induction [102]. For example, pyocyanin plays an important role in regulating the synthesis and release of eDNA, and the inhibition of the synthesis of pyocyanin can reduce the amount of eDNA synthesis in *P. aeruginosa*, thereby affecting the formation of the biofilms [24]. More importantly, quorum-sensing inhibitors PasD, heat shock protein, and propolis have the characteristic of reducing eDNA release and biofilm formation [103,104,105]. eDNA and DNA fragments can be applied to treat a *P. aeruginosa* airway infection, which is a novel antibacterial strategy [106]. The potential implications of this antibacterial strategy need to be further discussed.

### 5.3. eDNA in Escherichia

Currently, five recognized species of *Escherichia* have been identified including *E. coli*, *Escherichia albertii*, *Escherichia blattae*, *Escherichia fergusonii,* and *Escherichia vulneris* [107,108,109,110,111]. Among them, *E. coli* are the most comprehensively studied model organisms [112]. *E. coli* are responsible for about 80% of the cases of urinary tract infections, sepsis, and meningitis [113,114]. eDNA is a crucial structural component of *E. coli* biofilms, and it is associated with an increased antibiotic resistance of *E. coli* and HGT [115,116]. In addition, eDNA also acts as a nutrient source for *E. coli* during starvation, and eDNA promotes *E. coli* aggregation as a molecular bridge in a concentration- and length-dependent manner [60,117]. By acting as a danger signal and mimicking the host DNA, the curli/eDNA complexes formed from the interaction between the curli (a bacterial amyloid generated by *E. coli*) and the eDNA can trigger autoimmunity, thus, causing systemic lupus erythematosus (SLE) disease, and the antibodies against the curli/eDNA complex serve as markers for the exposure to bacterial products in SLE patients [118,119].

The synthesis and release of eDNA are affected by various factors. During the static growth of *E. coli*, the secretion of eDNA is regulated by the global regulator H-NS and the lipoprotein NlpI [120]. In the absence of antibiotic stress, the toxin gene *hipA* in toxin-antitoxin system can trigger cell death, thus, increasing the amount of eDNA in the biofilms [121]. Researchers have found that eDNA release can lead to formate accumulation in *E. coli* cultures, thereby inhibiting the mass transfer between the medium and cells during industrial production [122]. Camelliagenin can interact with MDH and eDNA, thus, inhibiting the formation of the bacterial biofilm, and thus, amelliagenin is expected to become a new generation of antibacterial drug. These findings of eDNA provide new perspective for dealing with antibiotic-resistant bacteria and in treating diseases [123].

### 5.4. eDNA in Other Microorganisms

In *Staphylococcus epidermidis*, the eDNA enhances initial cell adhesion and surface aggregation [45]. Additionally, in *Neisseria gonorrhoeae*, the single-stranded eDNA is of considerable importance in initial biofilm formation [124]. In *Acidovorax temperans*, the eDNA can meditate the attachment to glass wool [125]. In addition, the eDNA increases the biofilm viscoelasticity to protect the biofilms from mechanical and chemical stress by interacting with other matrix components in *Staphylococcus aureus*, *Staphylococcus epidermidis*, and *Streptococcus mutans* [126]. The eDNA release from *Enterococcus faecalis* is synergistically controlled by gelatinase and serine protease [22].

Interestingly, eDNA functions differently even among the same species. For example, eDNA is required to initiate the biofilm formation in the frequently carried pathogenic clonal complexes (CC) of *Neisseria meningitidis*, whereas an eDNA-independent pathway for biofilm formation was discovered in CC with a low point prevalence of *N. meningitidis* [16]. In *Caulobacter crescentus*, the eDNA can inhibit single cell attachment and biofilm formation [127]. In *Salmonella enterica* serovar Typhimurium and *S. enterica* serovar Typhi, a similar phenomenon has been observed, whereby a significant increase of biofilm formation was observed when DNaseI was added to conduct eDNA digestion [128].

## 6. Methods for eDNA Extraction and Quantification

In the past decades, several methods have been developed and improved for eDNA extraction and quantification from environmental samples, and great efforts have been devoted to increasing the eDNA extraction efficiency, and to reducing the iDNA contamination from cell lysis so as to reduce the underestimation or overestimation of the community diversity by a molecular analysis. However, so far there is no agreement on a standardized method. The method for eDNA extraction from different environments should meet the following criteria: (1) to achieve a high eDNA yield; (2) to be representative of the total eDNA of the samples; (3) to be suitable for the follow-up quantification and analysis.

It is not an easy task to extract eDNA from soil, sediments, and activated sludge since the eDNA is weakly or tightly adsorbed onto organic or inorganic particles [34]. A sodium phosphate (NaP) buffer was first applied to elute the eDNA off the solid phase due to the competition between the NaP buffer and the eDNA for binding sites on solid particles [129]. Therefore, this approach has been extensively employed with different modifications [18,72,130,131,132]. Since proteinase K can facilitates the desorption of eDNA, proteinase K is used to increase the yield of eDNA [133]. In addition, a new nuclease-based method has been developed for eDNA extraction from marine sediments, and this method utilizes the commercial nuclease to perform the hydrolysis of eDNA [134]. Another method for eDNA extraction is to use a TE (Tris-EDTA) buffer to separate the eDNA [5]. After the extraction, the crude eDNA extracts need to be further purified. There are multiple eDNA purification methods such as ethanol precipitation, BaSO_4_ precipitation, polyethylene glycol precipitation, chromatography, cetyltrimethylammonium bromide (CTAB) precipitation, and additionally, the DNA extraction kits are frequently used to purify the DNA (Figure 4) [1,5,18,129,135,136,137,138].

In liquid environments, eDNA extraction is not as difficult as it is in the solid phases. Usually, the water samples are firstly concentrated by a filter membrane. Various types of filter membranes have been used to extract the eDNA from liquid samples. In some commercial kit extraction, the silica gel membrane adsorptive method has been widely used to concentrate free DNA in an aqueous solution, but this method is only suitable for a small volume of water samples [139]. For a large volume of water samples, the mixed cellulose acetate and cellulose nitrate (MCE) is utilized to recover the spiked eDNA from 100 mL water, however, the eDNA recovery rate is only 16% [81]. A novel eDNA electrostatic adsorption method using nucleic acid adsorption particles (NAAPs, silica gel coated with Al(OH)_3_) is established to extract the eDNA from 10 L of water samples with an ARG recovery rate of up to 95% [140]. In addition, in the research on extracellular ARG propagation, the water samples are filtered through a 0.22-μm polyvinylidene fluoride filter (PVDF) [71]. A 0.22-μm Millipore Express Plus hydrophilic polyether sulfone (PES) membrane filter is applied to perform sequential filtration to separate the eARGs and the iARGs from the aquatic samples, thereby resulting a higher recovery rate of eDNA (ranging from 79.5% to 99%) [141]. The eDNA recovery rates from the environmental samples by using four different membrane filters including the PVDF filter, PES filter, nucleopore polycarbonate filter (PC), and MCE filter were compared, and the results showed that the MCE filter exhibited the best affinity performance for the eDNA, which was followed by the PVDF filter and the PES filter, and the PC filter displayed the lowest recovery rate [81]. Additionally, a new method using magnetic beads for eDNA extraction from water and sludge by a magnetic field was developed, and it turned out that this method could promote hydrogen binding with nucleotides [142].

It is noteworthy that wherever eDNA is extracted from solid or liquid samples, biases are introduced at each treatment step, unavoidably [143]. It has been observed that different extraction methods could lead to a species bias [144]. There is no one way to cover the entire microbial diversity. Thus, it is essential to determine the appropriate eDNA extraction method.

The quantitative assessment of eDNA is critical for an eDNA-based sequence analysis and a biodiversity analysis. The commonly used methods for eDNA quantification mainly include spectroscopy detection, fluorescence staining, and a quantitative PCR (Figure 5) [145]. The ultraviolet-visible spectroscopy is a traditional method for eDNA quantification, and the eDNA is considered as acceptable when the A260/A280 ratio is greater than or equal to 1.8 and the A260/A230 ratio is greater than or equal to 1.5 [129]. Apart from this method, the extracted eDNA is usually quantified by DNA-targeting fluorescence staining. For this purpose, a variety of fluorometric DNA dyes are used such as SYPO [12], PI [94], DDAO [12], DAPI [146], PicoGreen [62,147], SYBR Green [132], and ethidium bromide [94]. The fluorescence staining is frequently used to investigate the role of eDNA in biofilm formation [87,94]. The quantitative PCR method is also frequently used for eDNA quantification using specific target genes such as the prokaryotic 16S rRNA genes, eukaryotic 18S rRNA genes, and fungal internal transcribed spacer 1 (ITS) amplicons.

However, amplicon sequencing with these markers is more concerned about the composition of the microbial community, with the biological functions of microbial community being ignored. It is worth noting that a quantitative PCR is an inexpensive and rapid assay for DNA quantification, however, the PCR amplification process inevitably introduces some biases which could affect the microbial diversity [148]. Such PCR amplification biases are associated with primer specificity, genome size, rRNA gene copy number, template concentration, Taq polymerase misincorporation, and so on [148,149,150,151]. For example, primer specificity is an important issue, especially when one is trying to conduct the DNA quantification of homologous targeting sequences. It has been shown that the results of the microbial diversity are diverse with the use of different primers in the same sample [151]. Therefore, more efforts are needed to minimize the bias due to PCR amplification.

Furthermore, the results that have been obtained by different methods might be biased as well. For example, fluorescence spectroscopy detection for eDNA fraction accounted for 39–61% of the total DNA, while a qPCR revealed that the bacterial and archaeal eDNA accounted for 42–58% and 29–71% of it, respectively [134]. Additionally, in order to clarify the effect of eDNA on the community diversity, the photoreactive viability PCR indicator propidium monoazide (PMA) was used to remove the eDNA from the total DNA, and the results showed that the eDNA significantly affected the prokaryotic and fungal richness and the composition of the microbial communities [152].

In addition to the traditional methods for eDNA quantification that are mentioned above, the PCR-free approaches including hybridization capture sequencing and metagenomic sequencing have been developed for diversity analyses in recent years [153]. In the hybridization capture sequencing method, the eDNA extracts are taken as inputs for the library preparation, then, the human-designed probes which could hybridize with the targeting sequences are added to the library, and finally, the targeting sequences can be taken out from the library for sequencing [154]. Therefore, it allows for the generation of larger datasets for numerous samples. Yet, the biggest disadvantage of hybridization capture is the loss of the targeting sequences [155]. The research object of metagenomic methods is the genome combination of all of the community members [156]. Since there is no DNA enrichment step, the metagenomic approach can overcome the drawbacks and limitations of qPCR and hybridization capture sequencing, enabling it to be an unbiased approach of monitoring diversity. Metagenomic sequencing could be used to identify multiple organisms and their functions. For example, a metagenomic analysis of eDNA in complicated infection samples provides better viable microbiome profiles [157]. In addition, metagenomic sequencing was utilized to assess the distribution and movability of eARGs of the microbial hosts in activated sludge [158]. Later, a new unique workflow was developed for biomonitoring, employing the high-throughput sequencing of eDNA from water samples; the sequencing result showed the biodiversity from microorganisms to microorganisms’ with relatively low abundance [153].

## 7. Summary and Perspectives

Extracellular DNA accounts for a large proportion of the total DNA. The study of eDNA provides a new insight into microbial biodiversity in various ecological environments such as soil, sediments, and feces. Mounting evidence suggests that there are two different pathways of eDNA release including the lysis-independent pathway and the lysis-dependent pathway. Once the eDNA is released into the environments, eDNA can perform quite a lot of ecological functions. For example, it plays an essential role in multiple biological processes such as HGT, nutrition supply, as well as biofilm formation. Furthermore, eDNA-based biomonitoring has gradually emerged as a potent tool for detecting biodiversity. Therefore, these eDNA characteristics can be informative to provide deep insight into the fate of DNA in ecosystems.

All of these biological properties of eDNA make it applicable in various fields. Currently, eDNA extraction methods are still very limited. In order to make the eDNA extracts more representative, the eDNA extraction methods need to be further expanded and optimized. In addition, the existing eDNA quantification methods have their limitations. For example, when a qPCR is used for eDNA quantification, the microbial diversity may be overestimated due to a diverse, known technical bias such as PCR amplification bias and marker bias. The metagenomic-sequencing approach, a prospective analytical technique, is PCR-free and this could be used to evaluate and monitor the microbial diversity on a large scale. Considering this, a metagenomic approach may be a more effective one to obtain more comprehensive information through high-throughput sequencing.

## Figures and Tables

**Figure 1 ijms-23-13690-f001:**
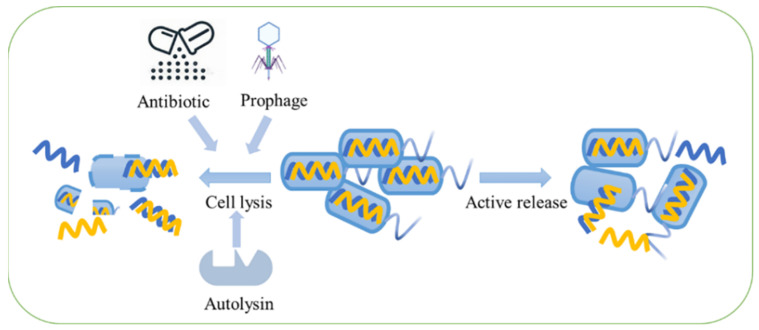
The origin of eDNA. Two pathways are included in eDNA release; one is the lysis-dependent way, and the other is the lysis-independent way.

**Figure 2 ijms-23-13690-f002:**
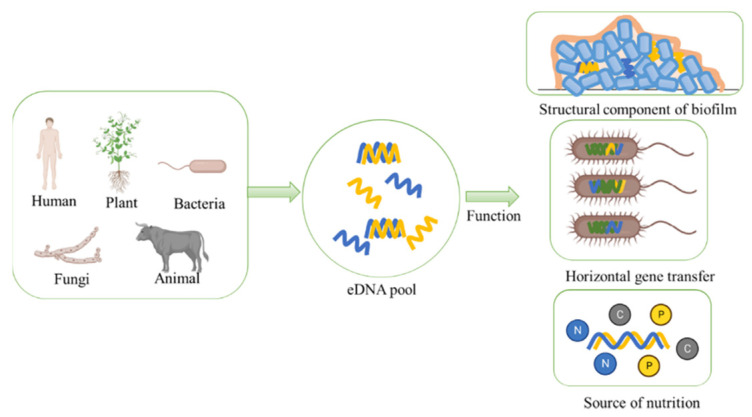
The origin and ecological functions of eDNA. eDNA could be obtained from different species such as human, plant, bacteria, fungi, and animal. The eDNA has three main ecological functions, firstly, it is a structural component of biofilm; secondly, it can take part in HGT; thirdly, it is a main compound in nutrient cycle.

**Figure 3 ijms-23-13690-f003:**
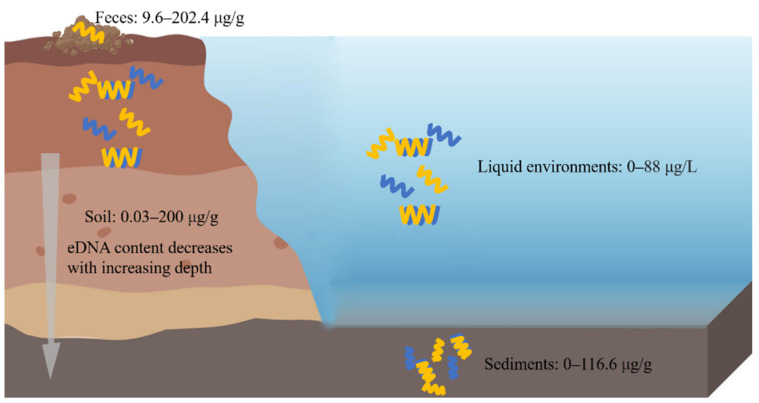
eDNA in different ecological environments. eDNA is widely distributed in soil, sediments, feces, and liquid environments. The left side of the picture represents the soil, and above the soil are the feces, and on the right side of the picture are the liquid environments, and under these is the sediment.

**Figure 4 ijms-23-13690-f004:**
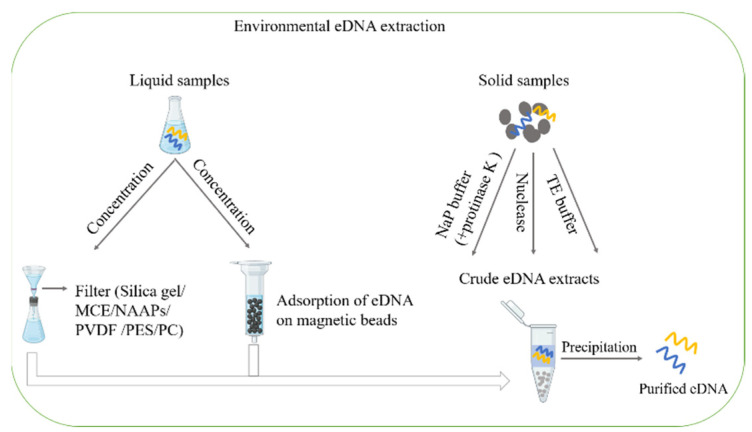
The methods of environmental eDNA extraction from liquid samples and solid samples. MCE: mixed cellulose acetate and cellulose nitrate; NAAPs: nucleic acid adsorption particles; PVDF: polyvinylidene fluoride filter; PES: polyether sulfone; PC: polycarbonate; NaP: sodium phosphate; TE: Tris EDTA.

**Figure 5 ijms-23-13690-f005:**
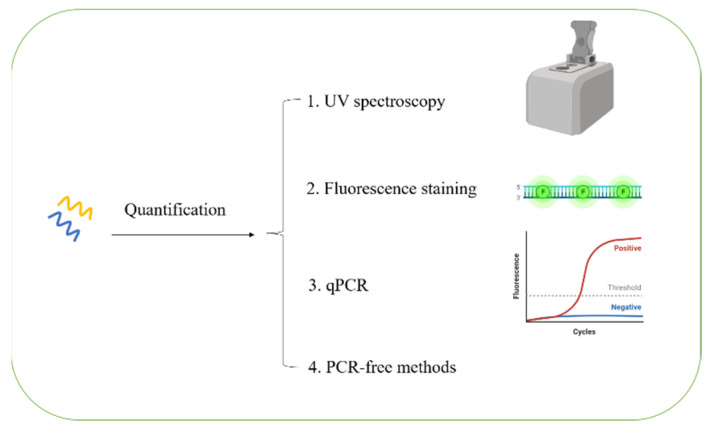
The methods of eDNA quantification. qPCR: quantitative PCR. There are four methods for eDNA quantification. Instrument in the upper right corner means the UV spectrophotometer for quantifying eDNA. The green circle represents fluorescent dyes.

## Data Availability

Not applicable.

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
