# Peer review of "The Origin, Function, Distribution, Quantification, and Research Advances of Extracellular DNA"

_ijms, 2022, doi:10.3390/ijms232213690_

Round 1

Reviewer 1 Report

The paper "A review on extracellular DNA: Origin, Functions, Distribution, Research Advances in Model Organisms" is dealing with a subject that has already been similarly reviewed. For example the review by Nagler et al. (2018) "Extracellular DNA in natural environments: features, relevance and applications". Applied Microbiology and Biotechnology (2018) 102:6343–6356. https://doi.org/10.1007/s00253-018-9120-4 is missing in the bibliography but should be included and commented given the similar topic. However, despite having noticed this, I still think that the work is nicely presented and may add some information on a quite relevant subject.

I find a problem with the title that is completely misleading when referring in general to Model Organisms. Indeed, after a general overview there is a main focus on tree bacteria and then some reference on other microrganisms. Reading the title a reader would expect a much broader presentation to "model organisms" which should then include eukaria as well. I strongly suggest to change the title downgrading the statement only to "Research advances in bacteria". 

Moreover, the review presents a paragraph on Methods for eDNA extraction and quantification that is interesting and represents a novel subject compared to existing reviews. Then mentioning issues on methods would be useful in the title.

In the final conclusion, the authors states that "microbial diversity may be overestimated due to PCR amplification bias". I find this a very interesting methodological issue that should find a better explanation in the previous section. Perhaps the author could elaborate a specific paragraph on this problem and another on bioinformatic critical problems that also may create significan biases in eDNA analysis.

Reviewer 2 Report

This manuscript by Yang et al is a nice review on eDNA. The content is well-organized and clearly outlined. My only suggestion for further improving this manuscript is to expand/elaborate on recent advances in metagenomic approaches regarding the eDNA research. I think this aspect bears contemporary importance but is only skimmed through in the last sentence. 

Minor comments:

L95-98: How does the fact that DNaseI or proteinase digest EPS components establish the conclusion that eDNA interacts with proteins in the biofilm? Please elaborate. Based on the current observation it can only mean that both eDNA and protein are integral to the structure of biofilm.

L184: delete one of two "since".

To be consistent, Figure 3 should be moved to the end of section 2, after 3.3.

Reviewer 3 Report

This is an interesting review that highlights recent findings and speculations on the eDNA. My major concern is that it lacks updated information on the analysis of eDNA as a tool to assess the physiological status or organisms and in biomonitoring applications in ecology. In addition more details about the physical features of eDNA should be provided (i.e. average size, ssDNA vs dsDNA etc..).

Minor concerns:

1.    There are also some typos and grammar errors scattered across the text that should be fixed (i.e line 40, “so far” is repeated twice…line 93 “begin” should be “begins” etc…).

2.    Line 104. The sentence “one of mobile genetic elements” is too general. The authors should add more details on what types of elements they refer to.

3.    Line 126. The authors should rephrase “…can “eat” DNA” which is as an inappropriate scientific terminology.

Round 2

Reviewer 1 Report

The new revised manuscript is improved compared to the original version and includes my suggestions and annotations.

In its present form I consider it acceptable for publication after the following minor modification:

At line 30-31 in the Introduction, the sentence "And it has been found that 30 most eDNA has higher molecular weight (≥600 bp) than iDNA[5]" it is wrong. It is not true that eDNA is bigger than iDNA and the reference [5] does not contain such indication. Indeed this work reports on methods of separation between eDNA and iDNA without comparison on their size.

My suggestion is to delete this sentence from the Introduction but leaving the citation of reference [5] together with the references 2-4 of the previous paragraph.

Reviewer 3 Report

I am fine with the authors' response.

Author Response

Thank you for your review.